**communications** engineering

# Signal improved ultra-fast light-sheet microscope for large tissue imaging
Md Nasful Huda Prince [1], Benjamin Garcia[2], Cory Henn [2], Yating Yi[3], Etsuo A. Susaki[4], Yuki Watakabe [5,6], Tomomi Nemoto [5,6], Keith A. Lidke [1], Hu Zhao[3], Irene Salinas Remiro [2], Sheng Liu [1] & Tonmoy Chakraborty [1,7] ✉

Axially swept light-sheet microscope in conjunction with tissue clearing enables three-dimensional morphological investigation of millimeter-scaled tissues at isotropic sub-micron resolution. However, these microscopes suffer from low detection signal and slow imaging speed. Here we report a simple and efficient imaging platform that employs precise control of two fixed distant light-sheet foci for axial sweeping. This enables full field of view imaging at 40 frames per second, a four-fold improvement over the current state-of-the-art. In addition, in a particular frame rate, our method doubles the signal compared to the existing techniques. To augment the overall imaging performance, we also developed a deep learning based tissue information classifier that enables faster determination of tissue boundary. We demonstrated the performance of our imaging platform on various cleared tissue samples and delineated its robustness over a wide range of clearing protocols.

Light-sheet microscopy (LSM)[1] along with advances in tissue clearing techniques[2,3] is revolutionizing the field of large tissue imaging. Due to recent technological advances in tissue clearing, anatomical and morphological investigation can now be performed at any scale (sub-cellular to organ level) and almost any tissue. To minimize the scattering of light, most clearing techniques improve light penetration by homogenizing the refractive indices (RI) mismatch that exists within the tissue. As a result, more laboratories are now incorporating rapid 3D imaging and histological studies for model organoids or, organisms like mice[4,5], rats[6], rabbits[7], fruit flies[8], and even humans[9,10] into their bio-imaging workflows. Regardless of these advancements, imaging tissue at high resolution has its own challenges. Specifically, traditional imaging modalities like confocal[11–13] and 2-photon microscopy[14–17] are not suitable for large tissue imaging, owing to their slow imaging speed, high light dosage, poor penetration depth and anisotropic resolution. In contrast, LSMs, because of their high speed and optical sectioning, are becoming the method of choice to image large samples.

Various light-sheet microscopes dedicated to cleared tissue imaging have recently been developed. For example, CLARITY-optimized light-sheet microscopy (COLM)[18] combines the LSM with a specific tissue-clearing technique for large tissue imaging. Spherical-aberration-assisted Extended Depth-of-field (SPED) LSM[19], eliminates objective scan by

increasing the depth of the field of the detection objective through spherical aberration and achieves 12 volumes per second imaging of the whole larval zebrafish brain. However, both COLM and SPED are based on traditional LSM and dependent on the RI of the cleared tissue, achieving at cellular resolution. The ultramicroscopy technique[8,20] utilizes aspherical optics to generate a thin light sheet up to 3 mm long and when combed with multiview imaging techniques, is able to image millimeter-size samples at high speed and isotropic resolution. However, its resolution is limited to 3 μm and multiview image-fusion introduces additional complexity to the system. Light-sheet theta microscopy (LSTM)[9] employs two identical illumination arms arranged symmetrically with the detection objective at a non-orthogonal angle. By synchronizing a two-axes light-sheet (LS) translation with the camera's rolling shutter, it achieves large field of view (FOV) imaging with uniform resolution. However, due to the geometry constraint of the three objectives, the axial resolution of LSTM is limited to ~5 μm. Open-top light-sheet microscopes (OTLS) allow easy sample mounting and higher stability. Recent development of OTLS, by using a solid immersion meniscus lens (SIMlens)[21], reduces aberrations caused by the tilted arrangement of the detection objective to the sample holder. This enables further development of multi-immersion OTLS[10] and hybrid OTLS[4] that are compatible with a wide range of immersion medium of various RI.

[1]Department of Physics and Astronomy, University of New Mexico, Albuquerque, NM, USA. [2]Department of Biology, University of New Mexico, Albuquerque, NM, USA. [3]Chinese Institute for Brain Research, Beijing, China. [4]Department of Biochemistry and Systems Biomedicine, Graduate School of Medicine, Juntendo University, Tokyo, Japan. [5]Division of Biophotonics, National Institute for Physiological Sciences, National Institutes of Natural Sciences, 5-1 Higashiyama, Okazaki, Aichi, Japan. [6]Biophotonics Research Group, Exploratory Research Center for Life and Living Systems, National Institutes of Natural Sciences, 5-1 Higashiyama, Okazaki, Aichi, Japan. [7]Comprehensive Cancer Center, University of New Mexico, Albuquerque, NM, USA. ✉e-mail: tchakraborty@unm.edu

However, their axial resolution is limited to ~3 μm. Tiling LSM[22–24] utilizes a spatial light modulator (SLM) to scan the light sheet along its propagation direction in order to extend the FOV of a thin light sheet and achieve uniform resolution across large FOV. However, its axial resolution is again limited down to 1 μm.

Recently, axially swept light-sheet microscope (ASLM)[25–27] based cleared tissue imaging system, called ctASLM, demonstrated isotropic sub-micron resolution imaging of large tissues. In ASLM, an ultra-thin LS moves synchronously with the camera's rolling shutter[28] where only the waist of the LS is captured during scanning. As the LS waist is determined by the numerical aperture (NA) of the illumination objective and can be as thin as the lateral resolution of the detection objective, an isotropic sub-micron resolution can be achieved over a large FOV. By using multi-immersion objectives, ctASLM[26] demonstrated that it could maintain this feature over a broad range of immersion media that are often associated with various clearing protocols. To the best of our knowledge, ctASLM is by far the only LSM that demonstrated these features. As a result, several microscope modalities have recently used ASLM to develop their cleared tissue imaging system such as mesoSPIM[5], LSTM[9], open-top ASLM[29], etc. However, ASLM has two major limitations: (1) low signal efficiency as each row of pixels is exposed only for a fraction of the total exposure time; (2) limited imaging speed as the axial scanning of the LS over a large FOV behaves nonlinearly at high speed, which limits the frame rate to 10 frames per second (fps).

It should be noted here that while emerging multi-scale LSMs, like OTLS[4], can rapidly image selected regions at high speeds thereby reducing the overall imaging time, many biological applications require large tissues to be imaged at sub-micron resolution. For instance, in stem-cell research identifying the extremely rare (0.003%) interactions that happen between Hematopoietic Stem Cells (HSCs) and stromal cells in the bone marrow nich[26,30], in connectomics which involves comprehensively mapping the neuronal connections[31–33], 3D pathological analyses of whole biopsies and surgical specimens[34–36], requires imaging large volumes at sub-micron resolution. However, imaging such large specimens is prohibitively time-consuming using the current state-of-the-art which at 10 fps may take tens of hours.

To improve ASLM's performance in terms of speed and signal, as a proof-of-concept study, we previously developed a dual-foci illumination scheme where two light sheets are axially scanned with two synchronized rolling shutters of the sCMOS camera[37]. In that implementation, the distance between the two foci were controlled using two lens pairs. This study showed that it was possible to improve ASLM's performance by enhancing the signal or by doubling the acquisition frame rate. However, this design resulted in different magnifications between the two light-sheet generation arms, so that the foci separation changes during the axial scan, making synchronization difficult.

Here we demonstrate an unique optical design that alleviates current ASLM's limitations, thereby offering precise, sub-pixel, position-control of the two foci while synchronizing their movement with the camera rolling shutter. We call this system signal improved ultra-fast light-sheet microscope (SIFT). Since SIFT is built upon our previous work, ctASLM, it not only inherits its salient features, like isotropic resolution over a broad immersion media, but brings ASLM's performance to a new level by quadrupling its speed or doubling the signal. Since imaging large tissues can often take days, SIFT is transformative for the large tissue imaging community who will now be able to achieve their imaging goals in significantly less time. We augmented SIFT's performance by developing a GPU-enabled deep learning (DL) based classification network that distinguishes informative volumes from non-informative volumes, thereby reducing the time necessary to carry out mesoscale structural evaluation. Here, we present the resolution and speed enhancement of SIFT and demonstrate its application in imaging various tissue samples from different tissue-clearing protocols.

## Results
Owing to the sheer size of the specimens, cleared tissue imaging is often a time-consuming and data-intensive endeavor. As a result, there is usually a

lengthy wait period between successfully clearing a sample and having the data available for visualization and further analysis. Irrespective of the microscope used, often the specimen must go through several general steps, namely: mesoscale structural evaluation, high-resolution imaging, and stitching (Fig. 1a). Although different microscopy platforms might have more/fewer steps, all high-resolution tissues imaging must generally go through these three broad steps. Depending on the size and shape of the tissue, the time, complexity, and cost associated with the individual steps may be rate-limiting. This problem is intensified when using ASLM for sub-micron isotropic imaging, since due to the nature of its design, traditional ASLM-based microscopes are limited to 10 frames per second. This makes high-resolution imaging the slowest of the three steps. For example, using a camera exposure time of 100 ms the imaging time scales from a mere 26 min to 268 h when imaging a sample of 1 mm³ and 1 cm³, respectively (RI 1.52). There is an additional consideration that must be taken into account when imaging faster: decrease in signal. Although this is a general issue for all microscopes, for ASLM this is particularly problematic as each pixel is only illuminated for a fraction of the exposure time. Therefore, we set out to build a new microscope that could increase both the imaging speed and the signal performance of ASLM.

Our proof-of-concept study demonstrated that scanning with two light sheets, rather than one, improves both the signal strength and imaging speed[37]. This is primarily because for a particular frame rate, this method doubles the detection signal without requiring doubling the peak-illumination-power, thereby offering a gentler illumination scheme compared to the traditional single-focus ASLM. In addition, sweeping dual-foci over half of the FOV requires shorter travel for the linear focus actuator (LFA). This shorter travel eases the oscillatory motion of the LFA (Supplementary Fig. 1 and Supplementary Note 1). In spite of these potential advantages, we found that, maintaining the precise synchronization of the dual-foci with the two rolling shutters of the camera chip required that the two foci to be positioned with sub-pixel accuracy. This was difficult to realize in our previous setup, where the two foci are controlled with two lens pairs, thereby preventing us from achieving the true benefit of the dual-foci concept (Supplementary Fig. 2). Here, we addressed this challenge with our unique optical design (Fig. 1b).

### Simultaneous, synchronized and precise scanning of two light sheets
Here, we designed SIFT (depicted in Fig. 1b, Supplementary Figs. 3 and 4, and the equipment list in Supplementary Table 1) by splitting the traditional remote-focusing setup (shown as O1, O4 pair in Fig. 1b) into two stacked identical remote-focusing arms, as an intermediate step (Supplementary Fig. 5). These two remotes focusing arms, called arm1 and arm2 in Fig. 1c, are each pupil-matched to the original O1 and O4 using a precise lens-pair combination (Supplementary Note 2). This allowed us to maintain an identical path length (L) in each arm and satisfy the $4f$ configuration to avoid unequal effective-focal-lengths and magnification at the sample space (from illumination objective's point of view). Since pupil-matched remote focusing offers a linear range of motion using the movement of the mirror, the sub-pixel level motion of the foci was achieved by placing the mirrors on micrometer-controlled linear stages. An equal but opposite translation ($\pm \Delta z$) of the mirrors from the nominal-focus-position (NF) allowed us to symmetrically place the two foci from their NFs (Fig. 1c). For our case, this means that the two foci are always separated by 1024 pixels. The LFA synchronously moves the two foci along with the camera active pixels (rolling shutter) over the entire FOV, keeping the separation between them fixed. These two foci are then imaged by the detection objective (O5 in Fig. 1d) onto the sCMOS camera. We experimentally simulated this using a conventional achromatic lens where one can see these two separated foci (Fig. 1e and Supplementary Figs. 6 and 7). Using this experimental simulation, we created a 2D focus for axial sweeping to assess the performance of the coverage throughout the entire FOV. The precise control over the foci movement and the identical formation of the two foci enable SIFT to achieve uniform resolution over the entire FOV of 2048 × 2048 pixels (870 × 870 μm²), elucidating the sharp line even for 40 fps (Fig. 1e).

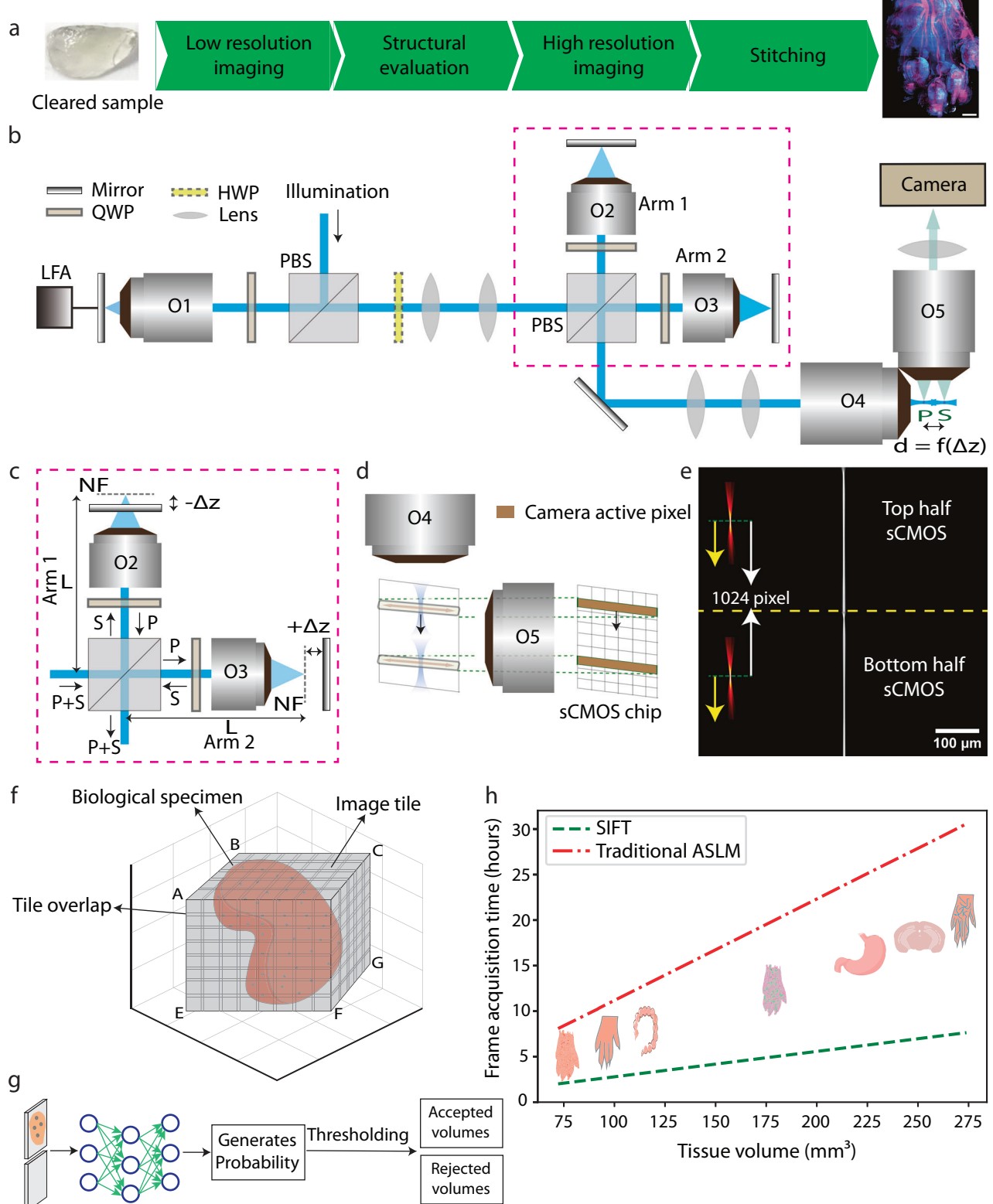

It must be emphasized that SIFT is not simply a faster ASLM. Although it may be possible to push the speed of ASLM to more than 10 fps imaging, using upcoming LFAs, the overall signal collected by the sCMOS is bound to decrease. This is simply because the active pixels of the rolling shutter will have an even smaller acquisition time. SIFT, on the other hand, offers the option to either improve the signal or the frame rate for a particular peak-illumination intensity while maintaining all the benefits of a traditional ASLM-based microscope. The signal improvement is realized in SIFT because, for a particular frame rate, each LS covers only half of the FOV. As a result, the effective exposure time at the active pixels doubles, which improves the acquired signal strength twofold. For samples that emit weak fluorescence signal, this could be a particularly useful feature since the users can double the signal without having to decrease the imaging speed. On the other hand, for specimens where the strength of the fluorescence signal is

**Fig. 1 | Pipeline using SIFT with isotropic imaging. a** Summary of the proposed imaging pipeline (from receiving the cleared tissue sample to visualizing the whole tissue image). **b** Schematic diagram depicting the stacked remote-refocusing units of SIFT. **c** Two, identical, remote-focusing arms split the incident light which is then reflected by a mirror that is $\Delta z$ distance away from the nominal focal plane of the objective. In each arm the $\Delta z$ is adjusted so that it results in two precisely separated foci in the sample space. **d** The synchronous movement of two foci with the camera rolling shutter. **e** To visualize our concept's effectiveness, we used two 2D foci (depicted by two yellow arrows towards left) to scan and synchronize with the rolling shutter feature of the sCMOS. Due to the precise control allowed by the stacked remote-focusing arms, high-quality synchronization was achieved for 25 ms camera exposure time over the entire $2048 \times 2048$ pixels ($870 \times 870$ m²) field of view (FOV), as depicted by the sharp line. **f** Since tissues come in all shapes and sizes, gauging the volumes for high-resolution imaging requires a mesoscale investigation, so that

unnecessary volumes can be removed. One way is to scan the tissue using a low-resolution scan that creates the outline of the tissue based on intensity-based classification. Depending on the shape and size of the tissue, even this step can take several hours. **g** The proposed pipeline introduces a DL-based binary classification network that distinguishes the informative volumes from the non-informative sets. **h** The frame acquisition timing comparison of SIFT with the traditional ASLM modality. At sub-micron isotropic resolution, SIFT can reduce the total frame acquisition time by at least fourfold compared to the traditional ASLM as evident by the slope of the frame acquisition time vs. volume ASLM (0.1113) and SIFT (0.0278). This does not include the time improvement offered by the DL-based classification. HWP half-wave plate, QWP quarter wave plate, LFA linear focus actuator, O objective, PBS polarizing beam splitter, NF nominal focus, sCMOS scientific complementary metal-oxide-semiconductor, SIFT signal improved ultra-fast light-sheet microscope, ASLM axially swept light-sheet microscope.

strong, for the same detection photon budget and illumination power, scanning dual-foci using SIFT will allow users to quadruple the frame rate relative to traditional ASLM. Not only is this more efficient in terms of time, money, and convenience, but for many clearing protocols (like BABB, 3DISCO, and CUBIC) the shelf-life cleared specimens is only a day or two long. This is because the fluorescence signal strength deteriorates over time. In these cases, rapid image acquisition is critical for imaging a sample in its entirety before signal is lost.

Because tissues have various shapes and sizes, gauging the accurate profile of tissue is a difficult task. Often, depending on the microscope modality being used, mesoscale structural evaluation may take several hours. This is an important step since an inaccurately assessed tissue map can cause unnecessary wastage of time (by imaging empty volumes at high resolution) and data storage. One common way to generate such tissue outlines is by assuming a cube encompassing the tissue and then carrying out rapid low-resolution imaging to identify those position coordinates which are relevant (have signals) (Fig. 1f). However, since these position coordinates can generally be in tens of thousands in number, assessment of these low-resolution tiles for structural evaluation is often a very slow process. Therefore, to supplement the improvement in imaging speed over traditional ASLM, here we designed SIFT to intelligently perform such mesoscale structural evaluation. To select tiles containing sufficient tissue information, we evaluate each tile with GPU-enabled DL-based classification algorithm (Fig. 1g) (details in "Methods"). The shortlisted positions were then imaged at high resolution and stitched together using BigStitcher[38] to reconstruct a large complex dataset of the whole tissue. We demonstrated the efficiency of our pipeline by imaging several specimens of different shapes and volumes (Supplementary Table 2). Due to the fourfold improvement of frame acquisition speed, SIFT reduced day-long image acquisition times to only few hours with a small compromise in signal strength. This faster imaging capability is evident by its four times smaller slope of frame acquisition time concerning the imaging volume compared to the traditional ASLM system across various specimens (Fig. 1h and Supplementary Fig. 8).

**Microscope quantification**

**Spatial resolution.** The microscope's spatial resolution performance is quantified by imaging 500 nm fluorescent beads embedded in 2% agarose and subsequently measuring the full-width half-maximum (FWHM) of the 3D point spread function (PSF). Stacks of fluorescent beads were acquired by axially moving the agarose cube, mounted on a custom holder, using a linear piezo stage. The axial step size is determined by the lateral pixel size at the image space. For instance, the measured magnification of SIFT in water is 15.28×, which gives a lateral image pixel size of 0.425 μm (Supplementary Table 3). The maximum intensity projection (MIP) of 40 planes both in lateral and axial directions show uniform resolution across the entire FOV (Fig. 2a and Supplementary Fig. 9). The measured FWHMs of several randomly selected beads in both lateral ($\sim 0.97 \pm 0.05$ μm, equivalent to 0.83 μm when RI is 1.56) ($n = 90$) and

axial direction ($\sim 0.97 \pm 0.06$ μm) ($n = 90$) delineates the isotropic resolution of SIFT (Fig. 2b and Supplementary Fig. 10). Richardson-Lucy iterative deconvolution sharpened the image by $\sim 17\%$, allowing SIFT to image at a FWHM of $\sim 0.80 \pm 0.06$ μm ($n = 75$) (Supplementary Figs. 11 and 12). We would like to mention here that, like other ASLMs, the images generated by SIFT are immediately available to users as 3D stacks "as-is" and do not require any computational post processing.

**Temporal resolution.** To quantify the temporal performance of SIFT, we reflect upon the two major portions of an ASLM cycle, namely, "exposure time" and "flyback time". As seen in Fig. 2c, d, an ASLM cycle is driven by a sawtooth waveform[27], where "exposure time" corresponds to the portion of the voltage that drives the LFA along with the synchronized rolling shutter of the sCMOS, while the "flyback time" also known as "settle time" is required for LFA to move back to its starting position. Imaging a single frame is dependent upon the camera exposure time. To quantify SIFT's temporal performance, we define temporal resolution as the time it takes to capture one camera frame at the maximum FOV. However, while acquiring a 3D stack, the entire cycle repeats itself depending on the number of frames in the stack. As a result, the "flyback time" also affects the stack acquisition time. Finally, imaging a large tissue requires acquisition of 3D stacks at multiple positions (several hundreds to thousands) to generate overlapping tiles. Usually, such position transfer has time delays, known as "stack delay", between two positions which is dominated by the response time of the 3D stage and the filter wheel, and the data writing. These response times, although system-dependent, along with the cycle time, decide the overall imaging time required to perform a multi-position acquisition. Therefore, here we termed the total time for multi-position tissue imaging as "total imaging time".

The imaging speed is purely dependent on a "mechanical" phenomenon, as in, the LFA remains synchronized with the sCMOS's rolling shutter. Ideally, if the mechanical motion of the LFA was linear at high speeds, the speed of ASLM should have been only limited to the speed of the rolling shutter. However, that is not the case, at least for the LFAs that have been used in ASLMs so far[25,26,37]. For ASLM, at high speeds (greater than 10 fps), the long ($\sim 1$ mm for traditional ASLM at this NA) mechanical travel of the LFA falls out of sync with the sCMOS's rolling shutter. We found that halving the stroke length actually reduces the burden on the LFA and makes the motion linear enough that it remains in sync with the rolling shutter up to 25 ms frame rate. This not only allows us to push the speed of ASLM to 4× faster but also opens up the possibility to explore newer LFA that are coming to the market (see "Discussion"). As can be seen in Fig. 2e and Supplementary Fig. 13, SIFT could acquire one frame covering the entire FOV ($870 \times 870$ μm²) within 25 ms while still maintaining tight synchronization between the rolling shutter and the moving LSs. In contrast, for a traditional ASLM this is limited to only 100 ms (Fig. 2d, f and Supplementary Fig. 2). This fourfold improvement in frame acquisition time is due to the novel optical design, which allowed the two foci to move synchronously over the entire FOV. Of note, we found that with around 10% sacrifice in the FOV,

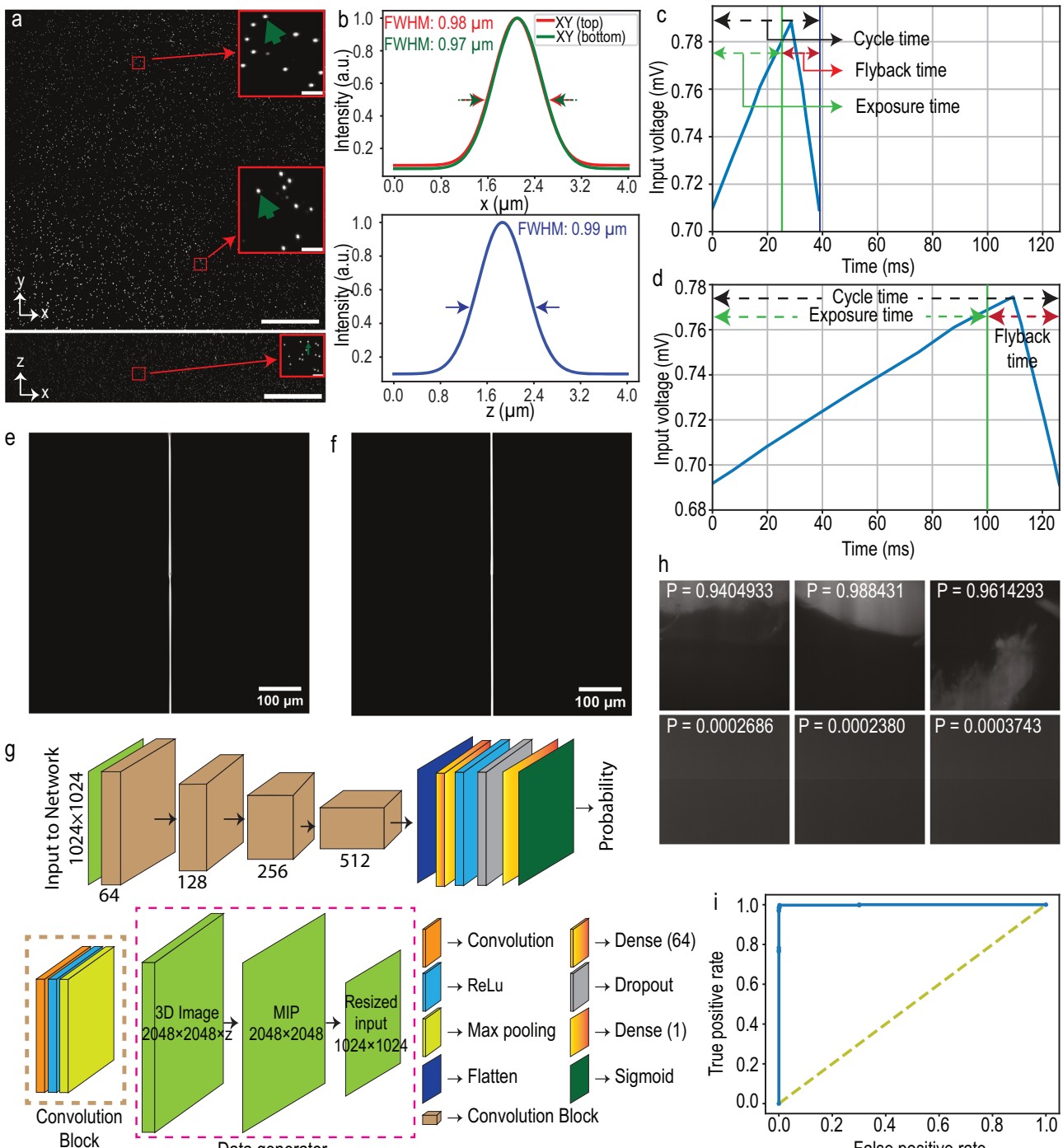

**Fig. 2 | Microscope quantification and DL network assessment. a** Maximum projection images of 500-nm fluorescent beads embedded in 2% agarose and submerged in water in XY and XZ, depicting uniform coverage over the entire FOV. The insets show the zoomed-in views of randomly chosen areas. **b** FWHM plots fitted with Gaussian profiles show isotropic resolution in all three dimensions. Deconvolution further improves the FWHM (Supplementary Fig. 11). **c, d** The sawtooth timing signal for LFA, synchronized with a succession of deterministic transistor-transistor logic (TTL) triggers for the camera and laser modulation for SIFT at 25 ms (**c**) and for the traditional ASLM at 100 ms (**d**) of camera exposure time. **e, f** A successive synchronization of 2D focus at the timing diagram shown in (**c, d**) with camera rolling shutter results sharp line across the entire FOV for SIFT at 25 ms (**e**)

and traditional ASLM at 100 ms (**f**) of camera exposure time. **g** DL-based binary classification network that distinguishes the informative images from the non-informative image sets. **h** Few representative probabilities generated by the DL network. Informative low-resolution images are labeled as one (shown in top row), whereas non-informative low-resolution images are labeled as zero (shown in bottom row). The trained network generates the probability map from where the image classes were distinguished by applying a threshold. **i** ROC curve of the classification network indicates a good classification performance of classifying the images from the validation dataset. Scale bars, 100 μm (**a**), 6 μm (inset of (**a**)). FWHM full-width at half-maximum, a.u. arbitrary unit, *P* probability, MIP maximum intensity projection.

SIFT could acquire one frame in even 20 ms of camera exposure time (Supplementary Fig. 6).

Next, we assess the performance of the Deep Learning (DL) based algorithm which we developed to assist in our mesoscale structural evaluation of the tissue boundary. DL is making its impact in various aspects of microscopy such as deconvolution[39], super-resolution image generation[40–43], classification[44,45], and segmentation[46,47]. Our DL-based classification model is able to distinguish the informative volumes from the non-informative volumes by generating a probability map of non-empty volumes with high accuracy (Fig. 2g). We found that this GPU-enabled DL-based classification is faster than an intensity-based approach ("Methods"). The trained DL network generates a classification probability of each low-resolution image tile (Fig. 2h). With a given discrimination threshold, the volume tiles were classified into two groups: informative and non-informative. The informative volume coordinate map generates the boundary of the tissue which is then fed to the microscope for high-resolution imaging. The receiver operating characteristic (ROC) curve delineates good performance of the trained network to classify the volume tiles (Fig. 2i) which corroborates the network efficiency (99.42%) found from the validation dataset.

## Large, volumetric, multi-color, high-resolution imaging of mouse forepaw, and single-color mouse stomach

To demonstrate how the per-frame improvement in the acquisition speed benefits the overall imaging time, we imaged several large, cleared tissue samples using SIFT and compared it to traditional ASLM imaging (Fig. 3a, b and Supplementary Figs. 14–24). For example, the PEGASOS cleared dual channel mouse forepaw (wnt1-Cre2, peripheral nerves and associated Schwann cells are labeled with R26-mScarlett^flox and imaged in the far-red fluorescence emission channel, and the green fluorescence emission channel is used for showing autofluorescence signal outlining gross structures) was imaged following the steps mentioned in Fig. 1a. In total, 3135 low-resolution image tiles were collected for single color which took 5 h and 11 min. Due to its irregular shape, approximately three-fourth of the image tiles did not contain tissue information, and only 836 image tiles were chosen by the DL network, which took around 20 min to process. The coordinate map of the informative image tiles was then fed to the microscope for high-resolution imaging. We imaged those 836 tiles for each channel (1672 tiles for dual channel) with a defined overlap between neighboring tiles in all three (XYZ) dimensions. The two channels

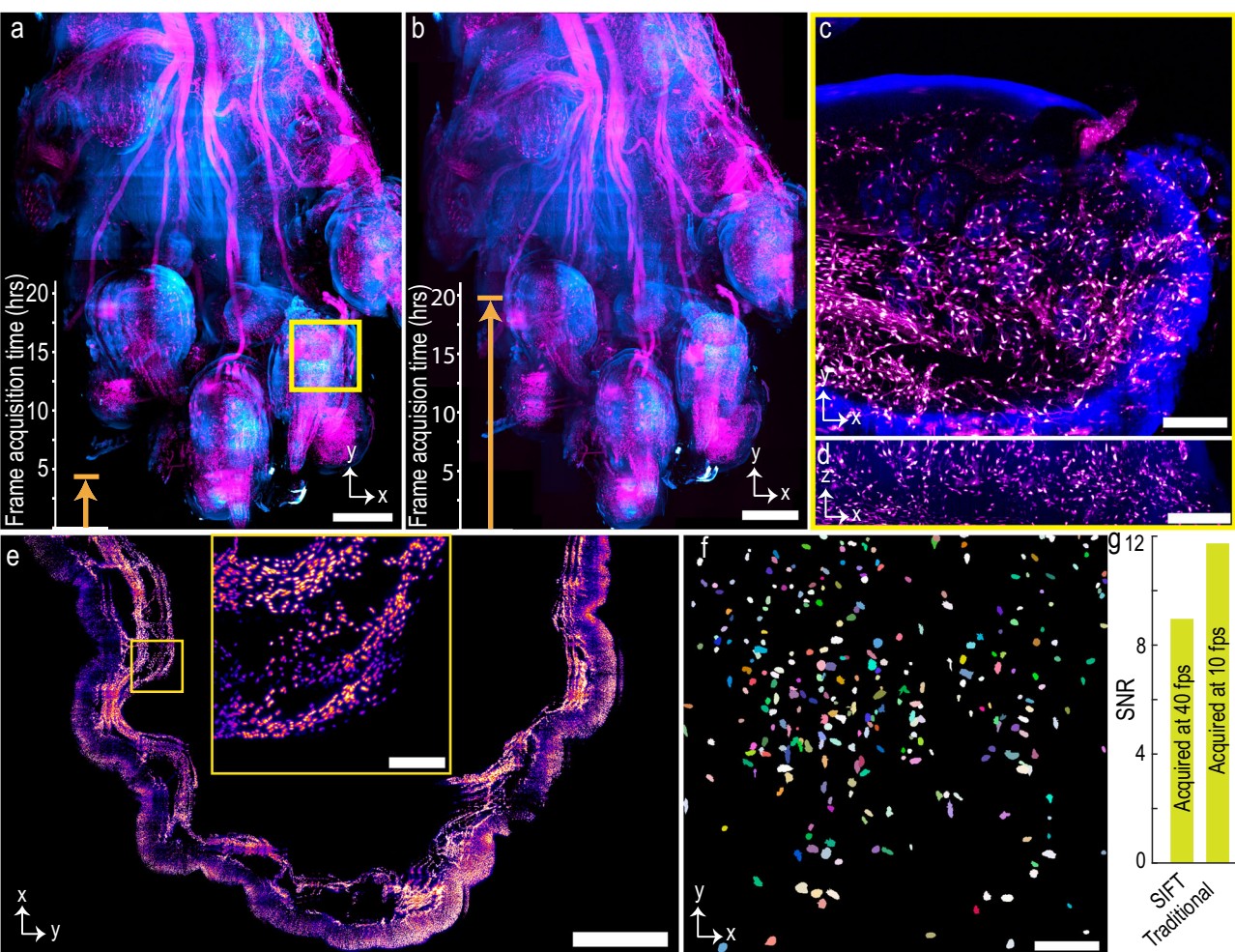

**Fig. 3 | Large volumetric imaging of mouse forepaw and stomach. a, b** Maximum intensity projection (MIP) of tissue images of forepaw, where the peripheral nerves and associated Schwann cells were labeled with R26-mScarlett^flox (magenta), and tissue structure is from autofluorescence (cyan), acquired by SIFT (**a**) and traditional ASLM (**b**) microscope. The image volume is 4.2 × 3.3 × 5.5 mm³. Total frame acquisition time for SIFT to image the tissue was 4.93 h, whereas the same for traditional ASLM was 19.73 h. **c, d** Lateral (**c**) and axial (**d**) view of a tile of a region of (**a**), marked by square yellow box. **e** MIP of Cdh5-cre Ail4 H2BGFP mouse stomach. The imaging volume was 4.4 × 7.1 × 3.2 mm³. The inset shows the higher magnification view of the selected region. **f** Segmented nuclei part of a randomly selected tile of mouse stomach, shown in (**e**). **g** Signal-to-noise ratio (SNR) plot of a randomly selected tile for both SIFT and traditional ASLM modality. The fourfold improvement of frame acquisition time compromises small SNR with respect to the traditional ASLM technique. The statistical SNR computation (Supplementary Fig. 25) for various tiles of two different samples verifies the generality of the signal strength of SIFT in comparison to traditional ASLM. Scale bars, 600 μm (**a**, **b**), 100 μm (**c**, **d**), 1 mm (**e**), 120 μm (inset of (**e**)), and 45 μm (**f**). Hrs hours, SNR signal-to-noise ratio, fps frames per second.

cumulatively generated 5.40 terabytes (TB) of image data which were then stitched using BigStitcher. The final stitched image with five times downsampling is 55.84 GB. The maximum intensity projection (MIP) of the stitched image, acquired by SIFT at 40 fps and traditional ASLM at 10 fps are shown in Fig. 3a (Supplementary Fig. 16) and Fig. 3b, respectively. We found that the total frame acquisition time for $4.2 \times 3.3 \times 5.5 \text{ mm}^3$ dual-color cleared mouse forepaw was 4.93 h while the same for traditional ASLM was 19.73 h. The orthogonal view of one single tile of the dual-color forepaw with fine detail was achieved due to the isotropic resolution of SIFT (Fig. 3c, d).

We also imaged a mouse stomach from a Cdh5-cre Ai14 H2BGFP transgenic line using SIFT at 40 fps (Supplementary Movie 1) and traditional ASLM at 10 fps. H2BGFP labels endothelium nuclei of the stomach. The tissue volume was $4.4 \times 7.1 \times 3.2 \text{ mm}^3$. With DL, 2171 image tiles (out of 5054 low-resolution tiles) were selected for high-resolution imaging. The total frame acquisition time to acquire 2171 image tiles (7.03 TB) for SIFT was 6.40 h, while the same was 25.63 h for the traditional ASLM microscope. The final stitched image, with six times downsampling, took 53.21 GB of storage. The MIP of 221 slices of the stitched image stack is shown in Fig. 3e. The inset shows the higher magnification view of the selected region (yellow box). The high-quality images obtained by SIFT allow us to segment individual endothelium nuclei (Fig. 3f). As is true for most microscopes, it is expected that the signal strength will degrade with an increase of the acquisition speed. Here, we computed the signal-to-noise ratio (SNR)[40,41] for SIFT and traditional ASLM. We found that even after decreasing the camera exposure time by 75% (at 25 ms) the SNR for SIFT decreased by only ~17% compared to the traditional ASLM (Fig. 3g) (statistical data for various tiles from different samples are shown in Supplementary Fig. 25).

## Multi-immersion imaging

The multi-immersion objective enables SIFT to image various samples cleared by different protocols and immersed in media with different RIs from 1.33 to 1.56 (Fig. 4). We imaged larval zebrafish 3 days post fertilization (dpf), immersed in water, RI ~1.33 (Fig. 4a). The *Tg(6xNFKB:EGFP)* zebrafish line produces EGFP upon binding of the nuclear factor-κB (NF-κB) promoter region, acting as a readout for inflammatory status[48]. NF-κB is a crucial transcription factor that plays a central role in regulating genes involved in inflammation, immune responses, and cell survival. Monitoring NF-κB expression through imaging allows researchers to visualize its dynamic activity patterns in developing zebrafish embryos, shedding light on the precise timing and spatial distribution of immune-related processes. This information is essential for understanding how NF-κB influences cell differentiation, tissue patterning, and the establishment of an effective immune defense system in the early stages of zebrafish development. Such studies not only contribute to our understanding of fundamental developmental biology but also offer potential insights into human health, as zebrafish serve as a valuable model for studying conserved molecular mechanisms that underlie both embryogenesis and immune responses. For imaging, larvae were anesthetized with tricaine and embedded in 1.5% agarose. We mounted the fish into a fluorinated ethylene propylene (FEP) tube which is RI matched with water (Supplementary Fig. 26). We next performed tail injuries as previously described to capture live pro-inflammatory cellular responses by tracking egfp+ cell migration over time[49]. We imaged the selected region of interest (ROI) at the tail injury site (yellow mark) every 15 min over the course of 16 h. The isotropic resolution of SIFT resolved the inflammatory response following tail injury from all three dimensions (Fig. 4b, Supplementary Fig. 27, and Supplementary Movies 2 and 3). We also investigated developmental changes in egfp+ cells under homeostatic conditions in the whole larva over 15 h of time. We imaged the whole fish every 30 min and captured staining patterns that agree with previous studies[48]. During this imaging window, we observed increase in nfkb expression in the stomach region, as well as sustained nfkb expression in cell clusters that appear to be neuromasts (Supplementary Fig. 28 and Supplementary Movie 4).

We also imaged proximal segments of C57BL/6J wild-type mouse colon cleared by ScaleCUBIC protocol[50–52] (RI ~1.48). Capturing dynamic processes and interactions occurring within the enteric nervous system (ENS), often referred to as 'second brain', and its microenvironment enables us studying the crucial role in regulating gastrointestinal functions. This includes investigating the crosstalk between enteric neurons, paneth cells, goblet cells, and immune cells such as macrophages and lymphocytes. By observing these interactions in real time and in situ, we can uncover insights into how the ENS communicates with immune cells to influence gut health, inflammation, and immune responses within the gastrointestinal tract. For imaging, the tissue was immunostained with anti-Tubulin beta 3 pan-neuronal marker (red). The labeled neurons in the submucosal and myenteric plexuses, with projections extending into the lamina propria are shown in Fig. 4c. The isotropic resolution of SIFT allows us to observe the detail in both lateral and axial dimension, where we capture dendrites and projections extending from the neuronal bodies (Fig. 4d, e, Supplementary Fig. 12, and Supplementary Movies 5 and 6).

We then imaged densely labeled Thy1-YFP-H mouse brain cleared by CUBIC-L/R protocol[53,54] (RI ~1.52) (Fig. 4f–i). We imaged the brain to visualize the environmental stimulated neuronal activities where we were able to unravel the activities of light-responsive neurons. The imaging volume is $1.7 \times 1.2 \times 0.343 \text{ cm}^3$. The number of tiles imaged were 4111 (each tile volume was $750 \times 750 \times 140 \text{ μm}^3$) to capture such large brain specimen which generated 11.8 TB of data. The lateral (XY) view of the mouse brain section allows us to resolve the YFP-expressing neuronal cells of the brain (Fig. 4f). We can even observe the neuronal dendrites with fine detail from a single tile in orthogonal views due to the isotropic imaging capability of SIFT (Fig. 4g–i). We also imaged another densely labeled Thy1 GFP mouse brain cleared by PEGASOS protocol (RI ~1.56) (Supplementary Fig. 29) (a section of nuclear-stained mouse brain is portrayed in Supplementary Fig. 30). It is worth mentioning that SIFT does not require any physical modification or realignment to image samples immersed in different media.

## Discussion

In this manuscript, we demonstrated a smart imaging pipeline to carry out isotropic sub-micron resolution, large FOV, cleared tissue imaging. We developed SIFT (Supplementary Note 3) utilizing a highly controlled dual-foci imaging scheme that improves the frame acquisition time at least fourfold compared to ctASLM. Our pipeline also introduces a DL-based classification network to reduce the imaging volume for high-resolution imaging. Our developed classifier is faster compared to an intensity-based algorithm (see "Methods"), and is agnostic to the specimen type, so no further training is required for new sample types. Besides the time efficiency of SIFT, it is impartial to the clearing protocols, i.e., no restriction to the refractive indices of the clearing solvent. Moreover, no physical realignment is required for imaging specimen cleared by different protocols. Although we add two additional RF objectives this makes optimization and troubleshooting easier. Finally, our cleared tissue imaging pipeline is successfully tested for various specimens where the pipeline reveals its robustness to achieve isotropic sub-micron resolution imaging for large tissue samples.

We believe that this improvement of overall imaging time while still maintaining isotropic sub-micron resolution will benefit the imaging community in several ways. For example, in a core facility a faster imaging modality may save few hundreds to thousands of dollars in imaging time on a single session (Supplementary Table 4). Moreover, it optimizes utilization of the microscope and other associated facilities along with human labor. More importantly, many clearing protocols, like 3DISCO[55], BABB[2], Fast 3D Clear[56], iDISCO[3] warrants a smaller turnaround time because of fluorophore quenching due to peroxide formation. A faster imaging platform ensures that these tissues have been imaged while they are at their peak performance. In addition to this, an excellent program detecting the tissue border reduces the data collection from tens of terabytes to a few terabytes by removing the non-informative tiles.

The current imaging speed of SIFT is limited by the camera. It is critical for SIFT's performance that we use two independent rolling shutters

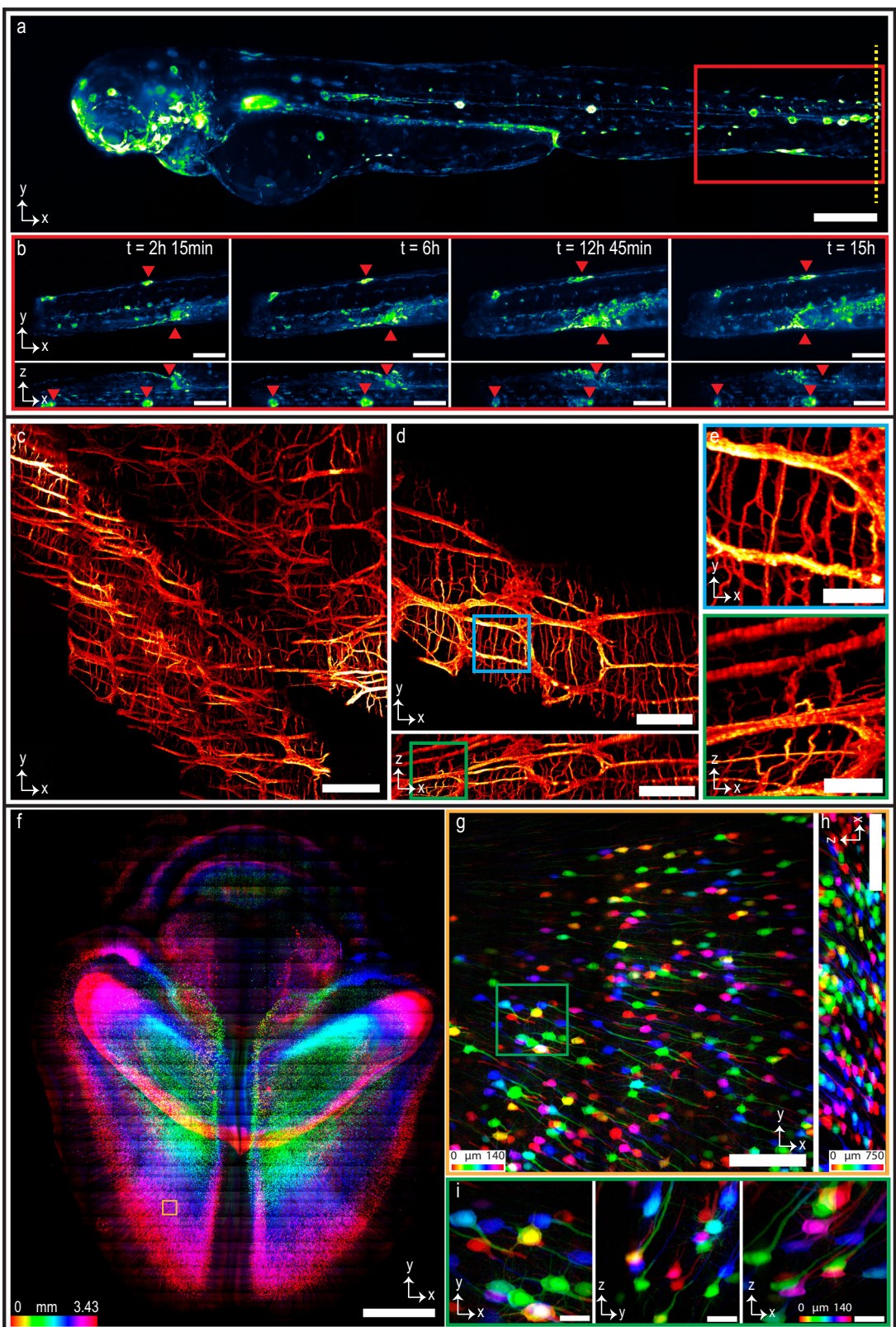

synchronized with two stacked LSs for the benefit of signal and speed enhancement. To the best of our knowledge, Hamamatsu Orca Flash 4 is the only sCMOS that currently offers this functionality. However, the pixel readout direction, for this camera, can be set in one direction only (top-to-bottom or, bottom-to-top). This poses a problem since, while acquiring a frame, the LFA is required to synchronously move with the direction of the rolling shutter and needs to return to its original position, at the end of the frame, so that the cycle can be repeated. This introduces almost a constant "flyback" time that cannot be used towards the actual imaging time. For smaller camera exposure time, this flyback time starts becoming comparable to the actual frame acquisition time and therefore the sawtooth waveform starts to take a triangular shape (Supplementary Fig. 31). Clearly, one

**Fig. 4 | Multi-immersion tissue imaging. a, b** Imaging genetically tagged zebrafish at RI ~1.33. The whole-body image of a larval *nfkb:egfp* zebrafish (3 dpf) (**a**). Time series of egfp⁺ cell movement (Supplementary Movie 2) of the location indicated by red box of (**a**) following tail injury (tail injury at the position shown with yellow dotted line in (**a, b**). Supplementary Fig. 27 and Supplementary Movie 3 delineate time series of egfp⁺ inflammatory cell movement of another depth stack. **c–e** Imaging the proximal segments of C57BL/6 J mouse colon at RI ~1.48 showing myenteric and sumucosal plexuses. MIP of the mouse colon cleared by CUBIC protocol displaying the neuronal populations of the myenteric and submucosal plexuses, with projections extending into the lamina propria (**c**). Isotropic resolution of SIFT allows us to visualize the orthogonal views of the mouse colon stained with anti-TUBB3 for a single image tile where we see both dendrites and projections of the neurons (**d**). Higher magnification of a myenteric plexus neuronal network corresponding to the selected region of (**d**) indicated by the square box (**e**). **f–i** Imaging mouse brain at RI ~1.52. MIP of the part of Thy1-YFP-H mouse brain section cleared by CUBIC-R + (N) (**f**). Lateral (**g**), and axial (**h**) view of a single tile of (**f**) (shown in brown square box) demonstrates the sufficient resolution of neuronal cell bodies and dendrites of the mouse brain. MIP of the higher magnification view in XYZ direction corresponding to the region of the stack shown by the square green box in (**g, i**). Scale bars, 200 μm (**a**), 100 μm (**b**), 180 μm (**c**), 120 μm (**d**), 40 μm (**e**), 2 mm (**f**), 160 μm (**g, h**), 30 μm (**i**).

potential way to reduce the "flyback time" is to use faster LFAs, like (THORLABS BLINK), which has faster response time compared to the voice coil that we used in SIFT. Alternatively, another approach is to modify the command signal to the LFA, which may be applied in real time, using existing techniques like *impulse-based* shaping[57,58] or commercially available tools like Input Shaping® to create a custom time domain input designed specifically to suppress vibrations created by motion transients. Another potential solution could be future sCMOS cameras with bidirectional dual-rolling shutter feature. This will ensure that the LFA does not have to return to its original position and frames can be acquired in "top-to-bottom" followed by 'bottom-to-top' mode, thereby obviating the "flyback" time altogether.

Besides the flyback delay, the stack delay (time delay between two consecutive stacks) also prevents us from getting the maximum time benefit that SIFT has to offer. Though SIFT can shorten the frame acquisition time by at least fourfold (Fig. 1h), both the flyback and stack delays cause the total imaging time longer than the total frame acquisition time, where the total imaging time for our system is improved by around threefold (Supplementary Fig. 32). For instance, although the total frame acquisition time of imaging 274 mm³ mouse hind paw for SIFT and traditional ASLM are 7.63 and 30.55 h, respectively, the total imaging time for SIFT and traditional ASLM are 13.91 and 40.48 h respectively. It is also worth noting that this stack delay is system-dependent and is dominated by the response time of the 3D linear stage and the filter wheel.

Finally, we want to emphasize that our assertion in this manuscript does not merely propose an arbitrary speed enhancement for ASLM; instead, we introduce a method that we believe opens up further opportunities for exploration. Regardless of the type of actuator employed, our approach offers an efficient technique in terms of signal and speed. Our unique stacked remote-focusing geometry allows us to divide the camera's FOV into two identical halves, with equal magnifications, while maintaining the critical *4 f* geometry integral to aberration-free ASLM's performance. Typically, one might assume that reducing the FOV in half would double the speed but we found that for mechanical actuators halving the FOV alleviates the actuator's workload, enabling a speed improvement of more than twofold (at least quadruple in our case). Given that our instrument is designed to accommodate all cleared tissue protocols, including live cell imaging, we believe that SIFT will be an invaluable addition to any biomedical imaging facility.

## Methods
### Optical setup
The SIFT modality is delineated in Supplementary Figs. 4 and 33 along with the list of the parts is tabulated in Supplementary Table 1. The optical setup is illustrated in three different sections. The first section is the beam accumulation, cleaning, and LS generation.

In the excitation path, four lasers at wavelengths of 405, 488, 561, and 637 (LX 405-100 C, LX 488-50 C, LS 561-50 and LX 637-140 C, Coherent OBIS) are combined through a series of dichroic mirrors (LM01-427-25, LM01-503-25 and LM01-613-25, Semrock). The combined beam is cleaned with a spatial filter (a 50 mm lens (AC254-50-A, Thorlabs) and a 30 μm pinhole (P30D, Thorlabs) and re-collimated by a 200-mm lens (AC254-200-A-ML, Thorlabs). The collimated beam is further expanded by a 5×

Galilean beam expander (GEB05-A), resulting in a 20-fold expansion from the initial laser beam. The expanded laser beam passes through a cylindrical lens (ACY254-50-A, Thorlabs) to generate a Gaussian LS. The LS is focused in × dimension at a resonant mirror galvanometer (CRS 4 kHz, Cambridge Technology) which dithers the LS along *y* axis to reduce the shadow effect from obstruct structures from the sample.

The second section is the light illumination optics constituted by two controlled foci capable to move keeping a fixed distance between them. The LS then passes through a 200 mm lens (AC508-200-A, ThorLabs) and is reflected by a polarization beam splitter (PBS) (10FC16PB.7, Newport). A half-wave plate (HWP) (AHWP3, ThorLabs) is used to maximize the reflection at the PBS. The laser beam then passes through a quarter wave plate (QWP) (AQWP3, ThorLabs), becoming circular polarized and focused to a tiny mirror by objective 1 (XL Fluor x4, NA 0.28, Olympus Life Sciences). The tiny mirror is mounted on a linear focus actuator (LFA, LFA-2010, Equipment Solutions), with 10 mm travel range, 50 nm repeatability and 3 millisecond response time (at maximum travel range). A N-BK7 optical window (37-005, Edmund Optics) is used between the tiny mirror and objective 1. The reflected light is then collected by objective 1 and passes through the QWP where the laser beam becomes P polarized and transmits through the PBS. The laser beam then passes through a pair of relay lenses and splits into two identical remote-focusing[59-61] arms through a second PBS. The two relay lenses, 200 mm (ACT508-200-A-ML, ThorLabs) and 75 mm (AC508-075-A-ML) form a 4 f system and conjugate the pupil of objective 1 to the pupil of the remote-focusing objective (Olympus UMP PlanFl 10×, NA 0.30)[62]. Each remote-focusing arm consists of a QWP, an objective and a mirror. The entire arm is mounted on a linear stage to fine control the path length. The mirror is also mounted on a linear stage to control the position of the LS focus. The focus separation between the two LSs are set to 1024 pixels. The reflected laser beam of each arm is collected by the objective and passes through the QWP with its polarization rotated by 90° and the two laser beams recombined at the output port of the PBS. The combined laser beam then passes through a pair of relay lenses, 200 mm (ACT508-200-A-ML, ThorLabs) and 150 mm (AC508-150-A-ML) which forms a 4 f system and conjugates the pupil of the remote-focusing objective to the pupil of the illumination objective (cleared tissue objective, Applied Scientific Instrumentation (ASI), Special optics 54-10-12). The laser beam is finally focused by the illumination objective to form two LSs.

In the detection path, fluorescence light from the sample is collected by the detection objective (cleared tissue objective, Applied Scientific Instrumentation (ASI), Special optics 54-10-12), passing through a tube lens (ITL200-A, ThorLabs), and forms images on a sCMOS camera (Orca Flash 4.0, Hamamatsu Corporation). The camera captured the images using "Dual light-sheet readout mode". Forty rows of readout strips were chosen active at a single instance to capture the light-sheet waist. A high-speed optical filter wheel (LAMBDA 10-B, Sutter Instrument) with three emission filters (FF01-525/30-25, FF01-605/15-25 and BLP01-647R-25, Semrock for green, red and far-red channel respectively) is installed before the camera for multi-color imaging.

Both the illumination and detection objectives are inserted into a cubic chamber with the front part immersed in the imaging media together with the cleared tissue sample. The imaging media is refractive index matched with the clearing protocol. The sample is mounted on a custom sample

holder and a three-dimensional motorized stage (Model: MP-285A, PCIe 80 7852 R) is used to translate the sample in 3D. During data acquisition, multiple tiles are collected across the sample by translating the motorized stage in a predefined cubical volume, a certain percentage of overlap in XYZ between neighboring tiles was used. Each tile was acquired by moving the stage axially with a step size equal to the pixel size for the corresponding immersion media.

## Microscope control and data acquisition

The microscope was controlled, and the images were acquired by a windows-operated Dell Precision 7920 computer, equipped with two Intel(R) Xenon(R) Silver 4210 R central processing units (CPUs) with clock speeds of 2.40 GHz and 2.39 GHz. The device is built with 128 GB of Memory, which is used to collect the microscopic data. The system additionally has an NVIDIA Quadro RTX 4000 Graphics processing unit (GPU), which has dedicated memory of 8 GB and shared memory of 63.8 GB (GPU memory: 71.71 GB). The system can run on a 64-bit operating system. It comprises with 2 sockets with 20 logical processors. The control software is based on LabView 2020 64-bit with Vision Development Module, and Labview FPGA Module. To actively modify with the scientific complementary metal-oxide semiconductor (sCMOS) camera (Flash 4.0, model: C13440-20CU) made by Hamamatsu, Japan, DCAM-API software was utilized for the Active Silicon Firebird frame-grabber. A 150 Watt Sutter instrument (100-240 V50/60 Hz; model: MP-285A) (PCIe 80 7852R, National Instruments) controls the 3D stage movement. A field programmable gate array (FPGA) produces deterministic transistor logic (TTL) trigger sequences and The produced triggers operate the resonant mirror galvanometers, stage positioning, voice coils, laser modulation and blanking and firing camera[37]. Engaging LFA with the system hardware is facilitated by the K-Hyper Terminal software. The Janelia Farms Research Campus of the Howard Hughes Medical Institute has licensed a few critical components as well as a few procedures under the agreement of material transfer.

## Sample preparation

**500 nm fluorescent beads**. In total, 11 ml fluorescent bead (F8813) stock dilution at 1:11 from the manufacture stock was prepared in a small bottle. Then, from the bead stock dilution 770-μl bead dilution at 1:11 was prepared in a centrifuge tube. The solution-containing centrifuge tube was sonicated twice for eight minutes each. In all, 50 μl bead dilution was taken and added to a new centrifuge tube, then vortexed for 3 min. In all, 800 μl of melted agarose (A9045-25G) gel was added to the centrifuge tube and vortexed for 10–15 s. The bead/agarose mixture was poured into a custom holder to solidify. The beads-in-agarose sample was then mounted for imaging.

**Animals**. For PEGASOS-cleared mouse: Following mice were purchased from the Jackson Lab with genotypes including *Thy1-EGFP-M* (JAX# 007788), *Ai14* (JAX# 007908), tTA^flox^ (JAX# 008600), tetO-H2BGFP (JAX# 005104) and *Wnt1-Cre2* (JAX# 022501). Tg(Cdh5-CreERT2) mice (Strain NO.T014691) were purchased from Gem-Pharmatech (Nanjing, China). R26-mScarlett^flox^ reporter was generated by Hu Zhao lab in the Chinese Institute for Brain Research. All animal experiments were approved by the Institutional Animal Care and Use Committee of the Chinese Institute for Brain Research. For tamoxifen treatment, tamoxifen (Sigma-Aldrich, T5648) was dissolved in corn oil (Sigma-Aldrich, C8267) at 20 mg/ml. The solution was kept at −20 °C and delivered via intraperitoneal injection or oral gavage for postnatal treatments.

For CUBIC-L/R cleared mouse: A fixed brain of 9-week-old female Thy1-YFP-H Tg mice (B6.Cg-Tg(Thy1-YFP)HJrs/J, The Jackson Laboratory, Identifier: 003782)[63] was provided from National Institute for Physiological Sciences (Okazaki, Japan) under the material transfer agreement with Juntendo University. All animal experiments were approved by Juntendo University (1569-2022279 and 1372-2022211), and National Institute for Physiological Sciences (22A044).

**PEGASOS-cleared samples**. Mouse forepaw, stomach and brain samples were processed following PEGASOS tissue-clearing protocol[64]. Briefly, mice were anesthetized with xylazine and ketamine. Transcardiac perfusion was performed with 50 ml ice-cold heparin PBS followed with 50 ml 4% PFA. Forepaws from *Wnt1-cre2; R26-mScarlett^flox^* mice, stomachs from *Cdh5-Cre^ERT2^; Ai14; tTA^flox^;tetO-H2B-GFP* (Endothelium dual-reporter or Endo-Dual) mice, and brains from *Thy1-EGFP* M mice were dissected and fixed at room temperature for 12 h, and washed with PBS at room temperature. For mouse forepaws, samples were immersed in 20% EDTA (pH 7.0) at 37 °C with shaking for 4 days and then washed with ddH$_2$O to remove residual salt. For stomach and brain samples, the decalcification step was skipped. Samples were then immersed in 25% Quadrol (Sigma-Aldrich, 122262) solution at 37 °C for decolorization for 2 days, with daily change of Quadrol solution. Subsequently, samples were immersed in gradient tert-butanol (tB, Sigma-Aldrich, 471712)) delipidation solutions for 1–2 days at 37 °C in a shaker, tB-PEG (containing 70% tB, 27% (v/v) poly (ethylene glycol) methyl ether methacrylate average Mn500 (Sigma-Aldrich, 447943) and 3% (w/v) Quadrol) at 37 °C with gentle shaking for 2 days for dehydration. Final clearing was achieved by immersing dehydrated samples in BB-PEG clearing medium (consisting of 75% (v/v) benzyl benzoate (Sigma-Aldrich, B6630), 22% (v/v) PEG MMA500 and 3% (w/v) Quadrol) at 37 °C until final transparency. Samples were kept in clearing medium before imaging.

**CUBIC cleared mouse colon**. On day 1, the proximal section of the colon was placed in a petri dish filled halfway with Hank's Balanced Salt Solution (HBSS) diluted to 1:10 in dH$_2$O and chilled to 4 °C. The petri dish was placed on a bed of ice to maintain temperature during processing. A 3-mL syringe was fitted with an oral gavage needle and filled with HBSS. The colon was held at one end with a tweezer, fully submerged in the HBSS, while the gavage needle was inserted into the lumen. The needle was pushed to the opposite end of the colon and HBSS was dispensed into the cavity as the needle was drawn backwards and out of the lumen. This was repeated as necessary to clear all visible debris out of the lumen. The tissue was then cut into ~1–2-cm sections. The sections were placed in 4% PFA and rotated overnight at room temperature. On day 2, the sections were washed two times for 2 h each with rotation with 1× PBS (0.01% sodium azide) solution. The sections were then immersed in 10 mL of ½ dH$_2$O diluted reagent 1 with rotation in a hybridization chamber at 37 °C with rotation overnight. On day 3, the sections were changed to 10 mL of Sca*l*eCUBIC-1 (reagent 1) and placed back on rotation at 37 °C. On day 5, the samples were washed with 1× PBS (0.01% sodium azide) solution three times for 1 hour each on a rotational agitator at 70 rpm. The samples were then blocked with 5% BSA in 1× PBS (0.01% sodium azide) at 37 °C with agitation for 3 h. Then the samples were incubated with anti-Tubulin beta 3 pan-neuronal antibody (1:300) in 1× PBS (0.01% sodium azide, 0.01% Tween, 5% BSA) with agitation at 37 °C overnight. On day 6, the samples were washed with PBS (0.01% sodium azide) three times for 1 h each at 70 rpm on the rotational agitator. Then the samples were incubated in AF588 anti-rabbit secondary (1:300) in 1× PBS (0.01% sodium azide, 0.01% Tween, 5% BSA) with agitation at 37 °C overnight. On day 7, the samples were washed with 1× PBS (0.01% sodium azide) solution three times for 1 hour each on a rotational agitator at 70 rpm. Then, the samples were placed in 10 mL of Sca*l*eCUBIC-2 (reagent 2) and agitated at 37 °C overnight. On day 8, the samples were placed in 10 mL of fresh reagent 2 and agitated gently at 37 °C until desired clearing was achieved.

**Live-imaging of zebrafish embryos**. *Tg(6xNFKB: EGFP)* embryos were collected and maintained at 28.5 °C at the UNM Biology Aquatic Animal Facility. Parents were maintained on a 14:10 light:dark photoperiod and fed a Gemma 300 diet (Skretting USA) twice per day. All parents were between 6 and 12 months of age. At 24 h post fertilization (hpf) embryos were moved from E3 media with 0.0001% Methylene blue to E3 media

containing 0.003% phenylthiourea to prevent the formation of melanin. At 48 hpf, embryos were dechorionated in a solution containing 1 mg/ml Pronase (Sigma-Aldrich Cat # 10165921001) for 5 min followed by three rinses for 5 min each in E3 media with 0.003% phenylthiourea. At 72 hpf, larvae were anesthetized in E3 media containing 200 mg/ml Tricaine (Syndel Cat #Tricaine1G) until response to physical touch was no longer present. For developmental imaging, larvae were placed into 1.5% low melt agarose at 42 °C, then drawn up into FEP tubing (ZEUS Virtual item: 0000183678). FEP tubing was then mounted in the imaging chamber, which was filled with E3 media with 200 mg/ml Tricaine. For imaging following injury, larvae were anesthetized as described above, and the posterior portion of the tail was cut with a sterile scalpel. Fish were then mounted as described above.

**CUBIC-L/R cleared mouse brain.** CUBIC-L/R tissue clearing was performed according to the previous literature[53,54]. In brief, a fixed whole brain of Thy1-YFP-H Tg mouse was treated with the commercialized CUBIC-L (Tokyo Chemical Industry, Japan, #T3740) at 37 °C for 4 days. The delipidated and PBS-washed sample was then stained with propidium iodide (PI) for counterstaining by immersing in HEPES-NaCl buffer (10 mM HEPES: TCI, #H0396; 500 mM NaCl: TCI, #S0572; 0.05% NaN$_3$: nacalai tesque, #31208-82) with 3 μg/mL PI (DOJINDO, Kumamoto, Japan, #343-07461) at 37 °C for 5 days. After PBS washing, the sample was RI matched with commercialized CUBIC-R + (N) (TCI #T3983) and then embedded in CUBIC-R-agarose as in the previous reports[54].

**Image stitching**
The open-source Bigstitcher[38] plugin for Fiji was used to stitch image tiles of large tissue. The image data was read from a local network connected with 10 GB data speed. Before stitching, the image tiles were converted and saved in a hierarchical data format (.h5) file using Bigstitcher. A 64-bit linux operating system-based machine with a 32 core in a single socket (AMD Ryzen Threadripper PRO 5975WX 32-Cores) was dedicatedly used for stitching of various datasets. Each core of the machine contains two virtual threads. The system is integrated with 512 GB memory with 3.6 GHz CPU speed and an A6000 Nvidia GPU. For example, stitching the datasets from imaging mouse stomach (2171 tiles, 7.03 TB of size, Fig. 3) required ~85 h and generated a stitched image with six times downsampling (53.2 GB) in.tif format.

**Deep learning-based classification network operation**
A deep learning (DL)-based classification network was developed and run on the same system used for stitching (Fig. 2). The model was made up with various convolution blocks and layers. Each convolution block extracted features from the images and max-pooled the image by reducing the shape by twofold. The training data included low-resolution informative volumes taken from various tissue samples and non-informative volumes taken from media-only sample. The informative and non-informative volumes are labeled as one and zero, respectively, and each type contains 2080 images (a total of 4160 images). 80% data were used for training, and the rest 20% are used for validation. The batch size used for the training is 32. An early stop algorithm is applied to monitor the validation accuracy with a patience time of 10 epochs. The training models at the checkpoints that improve the validation accuracy are saved. A.csv log is maintained to observe the training and validation accuracy. The best-trained model with accuracy more than 99% is used to classify low-resolution images. The probability of the volume being informative is output from the trained model (representative images are shown in Fig. 2). Given a threshold of the probability, the informative images are selected. The software is provided on the GitHub repository.

**Intensity-based classification**
We also developed an intensity-based classification algorithm to identify volumes with or without tissue information. The procedure is as follows: for each 3D image tile, (1) a MIP image along the axial dimension is generated, (2) the MIP image is smoothed by two uniform filters with a smooth kernel of $20 \times 20$ and $40 \times 40$ pixels respectively, (3) the difference of the two smoothed images is generated, this step is to remove the effect of background, (4) from the difference image, calculate the percentage of the pixels that are higher than a set threshold, here we use 2.5. (5) the percentage value is used as a score to classify whether the corresponding image tile contains tissue information, here we use a cut off value at 0.1%, where the image tiles with a score less than 0.1% are considered non-informative.

**Data analysis**
Images presented in this manuscript were not deconvolved. We showed that Richardson-lucy iterative deconvolution is able to further sharpen images acquired by SIFT (Supplementary Figs. 11 and 12). We used ChimeraX[65] for 3D volume rendering and 3D visualization. Stardist[66] plugin of Fiji was used to segment and analyze the nuclei (Fig. 3).

**Video rendering**
3D movie of tissue rendering was generated by ChimeraX[65]. Adobe premiere pro was used to assemble the rendered images into the final movie.

**Statistics and reproducibility**
In all, 500 nm fluorescence beads embedded in agarose gel were imaged ten times to quantify the spatial resolution of SIFT. Beads were also imaged using the conventional ASLM technique ten times for comparison purposes. FWHMs of the PSFs were measured from random selected beads within the FOV. Various large tissues, such as dual channel mouse forepaw (Fig. 3), dual channel mouse hind paw (Supplementary Fig. 11), mouse gut (Supplementary Fig. 14), mouse stomach (Supplementary Fig. 10) and whole mouse brain (Fig. 4 and Supplementary Figs. 15 and 29) were imaged using the same pipeline described in the main text. Each cleared tissue sample was imaged using both SIFT and traditional ASLM.

**Reporting summary**
Further information on research design is available in the Nature Portfolio Reporting Summary linked to this article.

## Data availability
The report and its Supplemental Information provide the primary results that underpin the findings of this study. The provided Github repository contains trained best checkpoint. The raw dataset used for this study is available upon request to the corresponding author.

## Code availability
All MATLAB and Python scripts needed for the pipeline are accessible at the GitHub repository[67]. The Python-based software includes the best-trained checkpoint, data generator, training and validation code, and the DL-based classification network.

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

## Acknowledgements

This project was supported by the National Cancer Institutes and the Cancer Center Support Grant through Grant Number P30CA118100, PRIME (AMED to E.A.S., grant number JP20gm6210027), JSPS KAKENHI grant-in-aid for scientific research (B) (to E.A.S., grant number 22H02824), Grants-in-Aid from the Takeda Science Foundation, Nakatani foundation for advancement of measuring technologies in biomedical engineering, Mochida Memorial Foundation for Medical and Pharmaceutical Research (to E.A.S.), and annual internal funding from the Chinese Institute for brain research. We would like to thank the University of New Mexico Start-up Grant for funding this research. This research was also supported by grants from NVIDIA and utilized an NVIDIA A6000 GPU. KAL was supported by NIH grant 1R01GM140284. This work was conducted with support from the University of New Mexico Office of the Vice President for Research Program for Enhancing Research Capacity. We would like to thank Dr. Kohei Otomo and Ms. Ayako Kato for helping sample preparation and shipment. We would also like to thank Dr. John Rawls for providing the zebrafish nfkb reporter line and the UNM Biology Aquatic Animal Facility staff.

## Author contributions

M.N.H.P. and T.C. designed the research. M.N.H.P. and T.C. designed, built and operated the microscope. M.N.H.P. performed image analysis, under guidance by S.L., K.A.L., and T.C. M.N.H.P., and B.G. imaged the zebrafish. M.N.H.P. and C.H. imaged the mouse colon. Y.Y. and H.Z. provided PEGASOS-cleared samples (mouse forepaw, hind paw, gut, stomach, brain); B.G., C.H., and I.S.R. provided specimens (zebrafish and CUBIC cleared mouse colon); E.A.S., Y.W., and N.T. provided CUBIC-R cleared Thy1-YFP-H Tg mouse brain; and guided for imaging those specimens. M.N.H.P., S.L., K.A.L., and T.C. wrote the manuscript. All authors read and provided feedback on the final manuscript.

## Competing interests

E.A.S. is a co-inventor on patents and patent applications owned by RIKEN covering the CUBIC reagents, and E.A.S. is employed by CUBICStars, Inc. that offers services based on CUBIC technology. The remaining authors declare no competing interests.
