## [Peer Review File · Communications Engineering]

This manuscript has been previously reviewed at another Nature Portfolio journal. This document only contains reviewer comments and rebuttal letters for versions considered at Communications Engineering.

Reviewer #1:

Remarks to the Author:

In this manuscript, Prince et al. develop a microscopy platform called “SIFT” in which two axially swept light sheets are aligned onto two separate halves of an Orca Flash 4.0 camera. This work builds upon the concept demonstrated in the author’s prior work in Dibaji et al. “Axial scanning of dual focus to improve light sheet microscopy” Biomedical Optics Express, 2022. Specifically, in this manuscript, the authors have redesigned the optical path from Dibaji et al. so that rather than using fixed lenses to generate the axially offset excitation beams, they use what is essentially an additional remote focusing unit after splitting by polarization, but using manually adjustable mirrors on this part rather than a high speed LFA. The key advantage of this approach over their previous paper is better magnification matching between the two arms thus enabling better synchronization with the rolling readout of the camera and a larger field of view. In the remainder of the paper, the authors show example data from a variety of samples imaged with this platform and also train a machine learning model to determine if an imaging tile contains enough signal to acquire at high resolution.

I believe that the manuscript and optical design are interesting and are suitable for publication in Communications Engineering, pending a few technical concerns described below.

Specific comments:

1) I agree with the authors that parallelizing the excitation in ASLM with two beams rather than a single beam will increase the duty cycle and lower the instantaneous intensity. The shorter scan range will reduce the demands on the remote focusing setup and increase imaging speed. The tradeoff for this parallelization is the potential for excitation bleed through wherein fluorescence photons excited by the defocused portion of the lightsheet on the other half of the camera will bleed into the readout strip on the opposite half. The authors address this in Figure 1E and in Figure S11. However, I’m concerned that these comparisons may be under reporting the amount of bleed through that will be present during imaging. This is because, in these figures, the authors do not dither the beam along the x-direction as they would normally during imaging. For a fixed beam like those shown, the intensity decays away from the beam focus in both the z and x directions. However, I anticipate that during scanning, there may be more bleed through as the beam is dithered in x since these emission photons excited by the broadened beam will add across all pixels of the readout stripe in x as the beam is scanned. This should be easy to test by filling the chamber with a fluorescent dye solution and counting the photons on each half of the camera when 1) only illuminating with the aligned beam, 2) only illuminating with the misaligned beam that is centered on the other camera half, and 3) illuminating with both beams simultaneously. These should all be done while dithering in x.

2) I believe there may be a typo on line 222: Is the field of view for SIFT really 870 mm?

3) I believe that the authors are using the Orca flash 4.0 in “Dual lightsheet readout mode”, but this isn’t

explicitly stated in the methods. If so, please include these settings and also list the number of active rows that were chosen for the readout stripe.

Reviewer's response from the 2nd review

Reviewer (Remarks to the Author):

In this manuscript, Prince et al. develop a microscopy platform called "SIFT" in which two axially swept light sheets are aligned onto two separate halves of an Orca Flash 4.0 camera. This work builds upon the concept demonstrated in the author's prior work in Dibaji et al. "Axial scanning of dual focus to improve light sheet microscopy" Biomedical Optics Express, 2022. Specifically, in this manuscript, the authors have redesigned the optical path from Dibaji et al. so that rather than using fixed lenses to generate the axially offset excitation beams, they use what is essentially an additional remote focusing unit after splitting by polarization, but using manually adjustable mirrors on this part rather than a high speed LFA. The key advantage of this approach over their previous paper is better magnification matching between the two arms thus enabling better synchronization with the rolling readout of the camera and a larger field of view. In the remainder of the paper, the authors show example data from a variety of samples imaged with this platform and also train a machine learning model to determine if an imaging tile contains enough signal to acquire at high resolution.

I believe that the manuscript and optical design are interesting and are suitable for publication in Communications Engineering, pending a few technical concerns described below.

Specific comments:

1. I agree with the authors that parallelizing the excitation in ASLM with two beams rather than a single beam will increase the duty cycle and lower the instantaneous intensity. The shorter scan range will reduce the demands on the remote focusing setup and increase imaging speed. The tradeoff for this parallelization is the potential for excitation bleed through wherein fluorescence photons excited by the defocused portion of the lightsheet on the other half of the camera will bleed into the readout strip on the opposite half. The authors address this in Figure 1E and in Figure S11. However, I'm concerned that these comparisons may be under reporting the amount of bleed through that will be present during imaging. This is because, in these figures, the authors do not dither the beam along the x-direction as they would normally during imaging. For a fixed beam like those shown, the intensity decays away from the beam focus in both the z and x directions. However, I anticipate that during scanning, there may be more bleed through as the beam is dithered in x since these emission photons excited by the broadened beam will add across all pixels of the readout stripe in x as the beam is scanned. This should be easy to test by filling the chamber with a fluorescent dye solution and counting the photons on each half of the camera when 1) only illuminating with the aligned beam, 2) only illuminating with the misaligned beam that is centered on the other camera half, and 3) illuminating with both beams simultaneously. These should all be done while dithering in x.

We are grateful for the reviewer's insightful observations. While we concur with the theoretical possibility of the dithered light sheet (LS) broadening and potentially increasing bleed-through, in practice we found this to be not concerning for our setup. The minimal dithering angle of 2.36 degrees, combined with a substantial separation of 435 μm between the two LSs, results in a negligible level of bleed-through.

In response to the reviewer's recommendation, we conducted an analysis using densely populated 500 nm fluorescent beads embedded in 2% low melting agarose. This setup enhanced our accuracy in identifying the LS focus during the dithering process, as depicted in **Figure R1**. Our methodology involved imaging using a Gaussian LS without Y-direction scanning. We then executed sequential imaging under three conditions: (1) with both LSs active, (2) with the top LS active and the bottom LS blocked, and (3) with the bottom LS active and the top LS blocked. According to the Reviewer's suggestion, these experiments were done by dithering the LSs. By consistently identifying the same group of beads, we were able to compare their maximum intensity profiles across these conditions (n=10).

The data presented in **Figure R1 a-c** clearly demonstrates that the waist of the Gaussian LSs remains distinct and well-separated, even when dithered. The individual beads' intensity profiles, shown in **Figure R1-f**, indicate a negligible bleed-through. Furthermore, a preliminary calculation regarding the LS waist and its expansion due to dithering (shown in **Figure R1-g**) reveals a mere 9 μm increase in one direction,

maintaining a separation of over 400 μm from its adjacent LS. SIFT in its design ensures that the separation of the LSs is always maintained during scanning which effectively minimizes bleed-through.

Consequently, we acknowledge that measuring bleed-through using a static (non-dithered) 2D focus could lead to underestimation. A representation like **Figure R1** might more accurately reflect such phenomena. We leave the decision to include this figure as supplementary material to the discretion of the editor.

Fig. R1 | Defocus bleed through measurement using 500 nm fluorescent beads. a-c, Maximum intensity projection (MIP) of 500 nm fluorescent beads embedded in 2% agarose and submerged in water while both LSs are active (a), only the top LS is active and the bottom one is blocked (b), only bottom LS is active and the top one is blocked (c). The

experiment was conducted for static Gaussian LS (not in ASLM mode i.e. no LS movement in Y) while the resonant galvo was dithering the LS in X direction. The red and green rectangle region delineates the waist of the corresponding LS. **d-e**, Zoomed-in view of a randomly selected region from around the edge of the X dimension of **b** and **c** (**d**), and **a** (**e**) where the bead from top and bottom provides approximately similar intensities. Beads from the top LS region of **a** and **b** are shown in the corresponding red and green square boxes (**e1** and **d1**) respectively, and the bottom LS region of **a** and **c** are shown in the corresponding red and green dotted square boxes (**e2** and **d2**). **f**, The maximum intensity of the beads taken from the region, shown in **d** and **e**, for each condition (**d1**, **d2**, **e1** and **e2**) delineates negligible bleed-through from one LS to another ($n = 10$). **g**, The LS was pivoted using a resonant galvo in X direction at 4 kHz frequency over a small angle (approximately 2.36°). This may cause the LS waist to broaden by about $9 \mu\text{m}$ leaving about $417 \mu\text{m}$ intermediate distance between the two LSs. Scale bars, $150 \mu\text{m}$ (**a,b,c**), $8 \mu\text{m}$ (**d,e**).

2. I believe there may be a typo on line 222: Is the field of view for SIFT really 870 mm?

We apologize for this typo. It is now changed to “ $870 \mu\text{m}$ ”.

3. I believe that the authors are using the Orca flash 4.0 in “Dual lightsheet readout mode”, but this isn’t explicitly stated in the methods. If so, please include these settings and also list the number of active rows that were chosen for the readout stripe.

We want to thank the reviewer for pointing this out. We now included the information about the light-sheet mode that we were operating and the number of active rows that were chosen (40 rows of pixels) for the readout stipe into the method section of our manuscript.

Reviewers' comments:

Reviewer #1 (Remarks to the Author):

I thank the authors for their response. I believe that the manuscript is suitable for publication.

I do have one minor comment regarding the rebuttal. In the first review, I had incorrectly assumed that the resonant mirror in Supplementary Figure 32 was pupil conjugate and served to dither the light sheet (as in digitally scanned light sheet implementations). I now see that the formation of the sheet is accomplished via lens CL1 and that the resonant galvo is sample conjugate to rotate the light sheet and reduce shadowing.

However given this, I'm confused how the authors generated the images in Supplementary Figures 2 and 11 and Main Figure 1E which shows what appears to be a radially symmetric Gaussian foci. Was the cylindrical lens removed or rotated for these two figures? Was it also removed for the bleedthrough quantification in Supplementary Figure 11? The intent of my prior comment in the original review was that I'd expect a greater amount of bleedthrough (as quantified via Supplementary Fig 11) for a sheet that is diverging only in "z" as opposed to the radially symmetric foci shown which diverge in both "x" and "z" as indicated on the graph. One minor note is that the Figure legend for Supplementary Figure 11 states "Light-sheets along with the vertical intensity profile" when the figure shows radially symmetric Gaussian beams rather than "sheets".

Reviewer's response from the 3rd review

Reviewer (Remarks to the Author):

I thank the authors for their response. I believe that the manuscript is suitable for publication. We thank the reviewer for this assessment. We feel that asking us to quantify the bleed-through using a light-sheet was indeed a very good suggestion and makes the manuscript more robust technically.

Specific comments:

1. I do have one minor comment regarding the rebuttal. In the first review, I had incorrectly assumed that the resonant mirror in Supplementary Figure 32 was pupil conjugate and served to dither the light sheet (as in digitally scanned light sheet implementations). I now see that the formation of the sheet is accomplished via lens CL1 and that the resonant galvo is sample conjugate to rotate the light sheet and reduce shadowing. We're pleased to see that the reviewer's confusion has been cleared up.

However given this, I'm confused how the authors generated the images in Supplementary Figures 2 and 11 and Main Figure 1E which shows what appears to be a radially symmetric Gaussian foci. Was the cylindrical lens removed or rotated for these two figures? Was it also removed for the bleedthrough quantification in Supplementary Figure 11? Yes, the cylindrical lens was removed and replaced with an equal focal length achromatic doublet to generate Supplementary Figures 2 and 11 and Main Figure 1E. The intent of my prior comment in the original review was that I'd expect a greater amount of bleedthrough (as quantified via Supplementary Fig 11) for a sheet that is diverging only in "z" as opposed to the radially symmetric foci shown which diverge in both "x" and "z" as indicated on the graph. We agree with the reviewer and therefore mentioned in our last rebuttal, that quantifying the bleed-through using a light-sheet, as it happens, is a better method than simulating the condition using a 2D foci. We have therefore included the quantification of bleed-through using light-sheet as supplementary (Supplementary Figure 33) and reference this in our main text. One minor note is that the Figure legend for Supplementary Figure 11 states "Light-sheets along with the vertical intensity profile" when the figure shows radially symmetric Gaussian beams rather than "sheets". We thank the reviewer for pointing this out. It was indeed a typo from our side. The legend is now corrected and replaced "Light-sheets" with "2D foci".